# Alcohol Brief Interventions (ABIs) for male remand prisoners: protocol for development of a complex intervention and feasibility study (PRISM-A)

Aisha Holloway,[1] Sarah Landale,[1] Jennifer Ferguson,[2] Dorothy Newbury-Birch,[2] Richard Parker,[3] Pam Smith,[1] Aziz Sheikh[4]

[1]Nursing Studies, School of Health in Social Science, University of Edinburgh, Edinburgh, UK
[2]Health and Social Care Institute, School of Health & Social Care, Teesside University, Middlesbrough, UK
[3]Usher Institute of Population Health Sciences and Informatics, University of Edinburgh, Edinburgh, UK
[4]Centre for Population Health Science, University of Edinburgh, Edinburgh, UK

**Correspondence to**
Professor Aisha Holloway;
Aisha.Holloway@ed.ac.uk

## ABSTRACT

**Introduction** In the UK, a significant proportion of male remand prisoners have alcohol problems. Alcohol Brief Interventions (ABIs) are an effective component of a population-level approach to harmful and hazardous drinking. ABIs have been shown to reduce the aggregate level of alcohol consumed and therefore reduce harm to the individual and to others. However, in relation to remand prisoners, there is no evidence as to how effective ABIs could be. The aims of this study are therefore to explore the feasibility and acceptability of an ABI for adult male remand prisoners and to develop an ABI for this group to be piloted in a future trial.

**Methods and analysis** The study will comprise three stages. Stage 1: a cross-sectional survey of adult male remand and convicted prisoners (n=500) at one Scottish prison and one English prison will be undertaken to assess acceptability and feasibility of delivering an ABI, as well as prevalence rates of harmful, hazardous and dependent drinking. Stage 2: in-depth interviews will be conducted with a sample of remand prisoners (n=24) who undertook the survey (n=12 in Scotland; n=12 in England). Two focus groups (one in Scotland and one in England) with six to eight key stakeholders associated with alcohol-related healthcare provision in prisons will be conducted to explore views on barriers, facilitators and levers to ABI delivery. Stage 3: through formal intervention mapping, the analysed data will inform the refinement of an acceptable ABI that is feasible to deliver to male remand prisoners.

**Ethics and dissemination** The project has been approved by the National Research Ethics Committee (NRES), National Offender Management System, Health Board Research and Development, Scottish Prison Service and ethics committee at The University of Edinburgh. Results will be published in peer-reviewed journals and presented at local, national and international conferences.

### Strengths and limitations of this study

► The first study to undertake Alcohol Brief Intervention development and feasibility in male remand prisoners.
► Prison staff and peer prisoners were involved in the recruitment process.
► Female remand prisoners were not recruited to the study.
► Those who could not understand or speak English to enable consent were unable to participate in the study and may limit the generalisability of the findings.

and social burden on individuals' families and society as a whole.[1] The evidence identifies a link between health and crime,[2] with a disproportionate level of health inequalities experienced by those individuals within the criminal justice system.

The delivery of Alcohol Brief Interventions (ABIs), as a method of addressing alcohol-related harm, was set by the Scottish Government in 2008 as a national Health Improvement, Efficiency, Access and Treatment (HEAT) target in three priority settings. The target evolved to become a HEAT standard for 2011–2013 and beyond, with National Health Service (NHS) boards and Alcohol and Drug Partnerships being responsible for its delivery in at least 90% of the priority settings of accident and emergency, primary care and antenatal care, with other settings accounting for the remaining 10%.[3] In the recent local delivery plan standard, priority settings will account for 80% of ABI delivery, with wider settings such as prisons comprising the remaining 20%.[4] Similarly, in England, Screening and Brief Interventions (SBIs) form part of The National Institute for Health and Care Excellence (NICE) quality standards[5] with the opportunistic delivery of

## INTRODUCTION

Harmful use of alcohol has been identified as a causal factor in more than 200 diseases and injuries, with alcohol contributing to 5.1% of the global burden of disease and injury, as measured by Disability-Adjusted Life Years (DALYS).[1] The impact of harmful alcohol can result in significant health, economic

BMJ

SBIs for adults drinking at harmful or hazardous levels as a role for health and social care staff.

## Background

There is robust evidence by way of systematic reviews and meta-analyses to indicate that ABIs are effective in reducing alcohol consumption among hazardous and harmful drinkers within healthcare settings.[6] A Cochrane collaboration systematic review of 29 primary care trials reported that ABIs delivered to patients were associated with a statistically significant reduction equivalent to four or five units a week at 1 year, in comparison to controls.[7] Nevertheless, there are weaknesses within the current evidence base: the majority of studies have been conducted in primary care; most trials have focused on middle-aged male drinkers; the optimal intensity and specific theoretical underpinnings of ABIs remains unclear.[7] The need for ABIs to be tailored specifically for the prison population is warranted to ensure they are relevant and acceptable to male remand prisoners.[8] A recent literature review of prison-based interventions identified 28 studies between 1995 and 2009. These were largely based in the USA with a focus on young offenders and women,[9] of which only one related to ABIs and targeted women prisoners. A more recent rapid systematic review identified no UK studies and only three from the USA of which none was remand focused.[10] Likewise, to our knowledge there are no evaluation studies of ABIs involving male remand prisoners in the UK.

Globally, there are over 10 million people incarcerated, with prisoners bearing a substantial burden of communicable and non-communicable diseases.[11] Since the 1940s, there has been an increasing trend in the number of individuals incarcerated in the UK. In England and Wales between 1990 and 2015, there was a rise in the prison population of just over 90%, with 64% in Scotland and in Northern Ireland the increase was 68% between 2000 and 2014/2015.[12] The total UK prison population in 2016 was just over 94 000 (95% male) with remand prisoners accounting for approximately 13%.[12] More recently, a study of prisoners identified that 70% surveyed, reported to having been under the influence of alcohol when committing the offence for which they were incarcerated.[8] The prevalence of alcohol problems in adult male remand prisoners is very high, with around three-quarters identified as having an alcohol use disorder (AUD) and around 40% with possible alcohol dependence.[8 13] Prison offers an opportunity for the identification, response and/or referral to treatment of those male remand prisoners who are consuming alcohol above recommended levels. Addressing alcohol harm in prisons can potentially reduce the risk of recidivism and reduce costs to society while tackling health inequalities.[14 15] Health savings of £4.3 million and crime savings of £100 million per year can be as a result of appropriate alcohol interventions.[16] ABIs have been associated with improved outcomes such as health utility EuroQol Five Dimensions Questionnaire (EQ-5D), service utilisation and reduction in alcohol-related harm[17] and are therefore of significant public health importance. However, access to 'mainstream' prison-based alcohol services such as alcohol screening, interventions, treatment and referral into additional services is typically not possible. This is due to the relatively short period of time remand prisoners are incarcerated, with those both unconvicted and convicted and unsentenced spending an average of 9 weeks in custody awaiting trial and/or sentencing. Approximately 25% of remand prisoners return to the community either as a result of being acquitted or receiving a non-custodial sentence. For these individuals, receiving an ABI as part of 'Through the Gate' or 'Healthcare Through Care'[18 19] could offer an important 'teachable moment', particularly if the ABI had a follow-on element. With the average prison sentence 16.2 months (56.8 months for indictable offences),[20] the impact of an ABI delivered to sentenced prisoners while previously on remand is less clear.

From the UK evidence, it is clear that those in contact with the criminal justice system are drinking at risky levels. In comparison to 25% of the general population,[21 22] we know that in the criminal justice system, 64% of young people, 53%–69% in the probation setting, 95% in the magistrate court setting, 64%–88% of adults in the police custody setting and 51%–83% in the prison system are classified as risky drinkers.[23] It is also noted that prisoner drinking norms differ widely to that of community consumption patterns.[24 25]

Alcohol-related crime in England and Wales is estimated to cost society £11 billion (2010–2011 costs). The association between alcohol use and crime is well documented.[26–29] Amount drank, pattern of drinking, context and individual elements have been identified as interactional influencing factors.[30] However, both long-term and short-term savings have been evidenced, with cost-effective early intervention to reduce alcohol use.[31]

There is limited evidence as to the effectiveness, optimum timing of delivery, recommended length, content, implementation, economic benefit and follow-up of ABIs in the prison setting for male remand prisoners.[10 32] This proposed early phase work is needed to adapt and develop a theoretically based self-efficacy enhancing ABI for use with male remand prisoners. There is also a need to establish whether such an intervention would be acceptable to male remand prisoners. Lack of sufficient early phase work risks an intervention that is poorly specified, lacks or has a weak, theoretical base and is less likely to deliver the desired outcomes.

This study will develop an ABI that is acceptable for delivery to male remand prisoners who have been identified as drinking alcohol at a level that is causing, or has caused, them harm (harmful or hazardous consumption) as defined by the WHO.[33] The study will also measure how feasible it is to deliver this intervention in the prison setting to male remand prisoners. The work is of importance to fill existing gaps in this area. We currently do not know what 'type' of ABI is needed or if it is acceptable to the male remand prisoner population. We are

also unsure as to the issues surrounding the feasibility of delivering such an intervention, and by whom, as well as follow-ups. This early phase study sets out to address these questions. The proposed ABI that will be developed from this early phase study will focus on enhancing self-efficacy with the aim of increasing drinking refusal self-efficacy. Self-efficacy derives from social cognitive theory[34] and has been identified as an important determinant of health behaviour, future health behaviour and health behaviour change. The four primary sources of self-efficacy information (that can be targeted) are performance attainment, vicarious experiences, verbal persuasion and physiological state.[34] Self-efficacy enhancing ABIs have been adapted in a range of health settings[35–37] and used by the principal investigator (PI) (AH) in two of these studies. The intervention here will be adapted and tailored for the unique circumstances of this group (male remand prisoners within a criminal justice setting) from an existing theoretically mapped self-efficacy enhancing ABI.[35 38 39] The development of an acceptable ABI will then enable us to undertake a future pilot to test the intervention, followed by a definitive randomised controlled trial (RCT) to evaluate efficacy and cost-effectiveness and ultimately an implementation study.

## Aim

The aims of this study are to explore the feasibility and acceptability of an ABI for adult male remand prisoners, to develop an appropriate ABI for adult male remand prisoners and a protocol for a multicentre randomised pilot study.

The specific objectives are:

▶ to identify the prevalence of self-reported hazardous/harmful alcohol consumption in adult male prisoners as identified by the Alcohol Use Disorders Identification Test (AUDIT) including reported views/personal experiences of ABIs delivered in prisons with acceptability of participation in a future ABI research study with follow-up;

▶ to explore adult male remand prisoners perspectives in relation to their beliefs and perceptions about their alcohol use, views regarding the acceptability of receiving an ABI while on remand, experiences of engaging with health professionals in prisons in relation to their alcohol use, the nature of this, the perceived impact and outcome of it and perceptions of acceptable alcohol screening, intervention delivery points and techniques, methods of delivery and by whom this should be delivered;

▶ to explore prison stakeholders' perceived feasibility and acceptability of an ABI for adult male remand prisoners among key stakeholders with a particular focus on their insights and experiences regarding the delivery of ABIs in prisons, perceptions of when, where and how in the current system, is the best place to screen and carry out ABI delivery and by whom, perceived mechanisms, processes, structures, training, cost, required to ensure feasibility of ABI

delivery from male remand prisoners, perceived views on resources and timing of delivery within existing workloads and priorities, perceived views on barriers, facilitators and levers to delivery (individual and organisational);

▶ to identify what an adapted intervention mapped self-efficacy enhancing ABI would comprise, based on the data collected and analysed in stage 1 and stage 2.

## METHODS AND ANALYSIS

The project will align to the early phase of the Medical Research Council's (MRC) framework for the development and evaluation of complex interventions.[40] Comprising three stages, a mixed-methods approach will be used with separate recruitment for prison stakeholders in stage 2. The data collection commenced June 2016 and will continue until January 2017.

### Participants and setting

For stage 1 (remand and non-remand) and stage 2 (remand prisoners) a purposive sample of male prisoners aged 18 and over who are currently imprisoned/detained within one Scottish prison within the Scottish Prison Service (SPS) prison estates and one English prison within the National Offender Management System (NOMS) will be recruited. To increase representativeness of prisons across the prison estate, we will aim to include two prisons with a range of prisoner categories and regimes. We will seek guidance from colleagues in criminal justice, NOMS and SPS to identify two appropriate prisons. Prison stakeholder participants for stage 2 will be those involved in the delivery of alcohol-related healthcare in prisons. They will be identified by the study advisory group as well as through existing networks.

#### Prison data collection preparation

We will work closely with prison staff to understand the daily routines, lockdowns and visiting times. In advance, we will also identify locations for interviews to take place and procedures for research assistants (RAs) to be escorted to minimise disruption and maximise recruitment. Following discussions with the prison sites, it is estimated that prisoner data collection will take 4 to 5 months.

#### Stage 1: prisoner participant inclusion criteria

We will recruit male remand and non-remand prisoners aged 18 years and over, who have been detained within one SPS Scottish prison study site and one NOMS English prison study site. They will have been incarcerated for 3 months or less and be willing to provide informed consent.

#### Stage 2: prisoner participant inclusion criteria

For stage 2, we will recruit male remand prisoners aged 18 years and over who have participated in the stage 1 survey and who have self-reported scores of 8 or over on the AUDIT[41] and willing to provide continued informed consent.

### Stages 1 and 2: prisoner participant exclusion criteria

Excluded individuals will comprise those unable to consent or deemed unable to make an informed decision regarding consent, and those considered by prison staff to be at risk of harm to self and to others. Specifically excluded will be those unable to give informed consent or deemed incompetent/unable to make an informed decision regarding consent, those posing a risk to self and/or others including on suicide risk management (act to care) or at risk due to being on any substance. Also excluded will be any prisoner subject to rule 41 or 95 (rules set out for the management of prisoners and young offender's institutions in Scotland)[42] or any prisoner on special security measures.

The prison staff will be trained by the researchers to understand the inclusion and exclusion criteria and will identify any inappropriate participants.

### Stage 2: stakeholder inclusion criteria

Involved in the delivery of alcohol-related healthcare in prisons (implementation, delivery, training, monitoring and/or commissioning) and willing to provide informed written consent.

### Stage 2: stakeholder exclusion criteria

Role does not involve the delivery of alcohol-related healthcare in prisons (implementation, delivery, training, monitoring and/or commissioning).

### Stage 1

#### Cross-sectional survey

##### Recruitment and consent

Prison staff and peer prisoners will provide potential participants with a short verbal account of the research study, together with a copy of the participant information sheet and a reply slip, in a sealable envelope during or after their prison induction. We will engage with service users through a community justice charity organisation when developing the study information leaflets and consent forms. The RAs will obtain informed written consent from potential participants. We will record the number of prisoners who are unable to consent due to language/literacy/cognitive impairment. For remand prisoners only, consent will also cover participation in both the survey (stage 1) and the in-depth interviews, if invited to take part in these (stage 2).

##### Study design

We will conduct an interviewer-led cross-sectional survey, delivered by the study RAs. Based on previous studies, we anticipate recruiting 500 participants (n=250 at each site) of that approximately 100 (n=50 at each site) will be male remand prisoners.[1 2] A sample size of 100 remand prisoners ensures that the half width of a two-sided 95% CI for a continuous outcome (eg, the total AUDIT score) is no more than 0.2 times the SD; and the half width of a two-sided 95% CI for a proportion (eg, the prevalence of drinking at harmful, hazardous and dependent levels) is no more than 10%. A sample size of 400 convicted

prisoners ensures that the half width of a two-sided 95% CI for a continuous outcome (eg, the total AUDIT score) is no more than 0.1 times the SD; and the half width of a two-sided 95% CI for a proportion (eg, the prevalence of drinking at harmful, hazardous and dependent levels) is no more than 5%.

Questions used in previous surveys will be adapted for use.[43 44] Basic demographic data, ethnicity, nature of charge and data from the 10-time screening instrument AUDIT will be recorded. The AUDIT is considered to be the gold standard for alcohol screening in healthcare settings.[41] The AUDIT can be scored between 0 and 40. A score of 8+ is referred to as a 'positive screen' and indicates an alcohol use disorder; hazardous drinking (score of 8–15), harmful drinking (16-19) or probable dependent drinking (20+). A score of 8 or more out of a possible 40 on the AUDIT is able to detect genuine excessive drinkers and to exclude false cases with sensitivity and specificity of 92% and 94%, respectively.[45]

To explore the acceptability of intervening, the questionnaire will ask participants how useful they would find a range of interventions, experiences of ever have receiving alcohol advice/information, their willingness to receive an intervention and whether in principle they would be willing to participate in an ABI research study with follow-up contact. Each interview survey document will have a unique study identifier (ID). The survey will be conducted where applicable privately in an identified meeting room within the prison. Questions will be read out to the participant and their answers recorded onto a hard copy of the survey.

##### Data entry and analysis

Quantitative data from the questionnaire will be coded and entered into SPSS software V.19.0. Data cleaning will be undertaken to identify and address any incomplete, incorrect or inaccurate data. The data will be cross-checked against the hard data. Descriptive statistics and 95% CIs will be used to summarise the data and inform the design of a potential future pilot trial. The quantitative analysis will be stratified such that the analysis will be conducted in the remand and non-remand prisoner groups separately. Variables will be classified and described using frequency tables, mean, median and SD. The prevalence of drinking at harmful, hazardous and dependent levels and corresponding 95% CIs will be calculated.

Open-ended responses from the survey will be coded into appropriate response categories. Where comments include multiple topics, we will code these into multiple categories. For those open-ended question responses that are illustrative, we will maintain these as verbatim responses to highlight key findings.

### Stage 2

#### Prisoner interviews and stakeholder focus groups

##### Prisoner interviews

*Study design*: In-depth face-to face interviews with a subsample of male remand prisoners who completed the stage 1 survey will be conducted. A sampling matrix will

be developed to inform a purposive sampling strategy to ensure representation in relation to range of AUDIT scores, age and previous experience of having an ABI. The RAs will liaise with the gatekeepers at each prison site to arrange appropriate dates and times for the in-depth interviews. For those who may have been released before a date and time has been arranged, there will be no further contact. A verbal recap with the participant information sheet will be undertaken by the RA and consent reconfirmed with opportunity for any questions to be answered.

Twelve male remand prisoners will be interviewed from each of the two study sites. Setting of the interviews will be as stage 1. The interviews will last approximately 45–60 min and will be digitally recorded at both study sites. Due to restrictions on the use of digital recorders, request and approval procedures for use of digital recorders will be sought at each study site. An interview schedule informed by the data collected from the stage 1 survey will guide the discussion. Topics will include participants' alcohol-related beliefs, experiences of engagement with health professionals or prison staff in relation to their alcohol use or any other individual while they have been in prison, perceived impact and outcome of this. We will also explore with them how best to maintain contact if they were to have participated in a follow-up trial. Each participant will also be shown an A4 page infographic outlining the key components and nature of an ABI. We will use this opportunity to explore their perceptions of the intervention content, intervention points and techniques to establish adaptations required to the existing self-efficacy enhancing ABI.

*Data entry and analysis*: The digitally recorded interviews will be anonymised and transcribed. Thematic analysis techniques using NViVo V.10 will be employed to produce initial codes categorising the content of each transcript. These codes will then be iteratively refined to produce emergent themes. Divergent and similar themes across interviews will be examined, comparing experiences and views regarding the acceptability, implementation, mechanisms, content and processes of an ABI and its delivery.

### Stakeholder focus groups

*Recruitment and consent*: Key stakeholders at each study site will be sent an introductory letter together with a participant information leaflet and consent form. A purposive sampling strategy will be adopted to ensure a range of professions, organisations and individuals are included. We anticipate that these are likely to include: prison nurses, commissioners, prison officers, prison health centre managers, external service provides, alcohol and drug partnerships in Scotland and equivalent in England. Where the stakeholder is an organisation, they will be asked to identify the most relevant individual to participate. The letter will ask that they contact the RA if they wish to participate and/or ask any questions that they may have about the study. The RA will gain written consent from those agreeing to participate prior to the focus group taking place.

*Study design*: Focus groups will take place with key stakeholders (n=6–8) from each study site and will be held in a location at a date and time convenient for participants. The RAs for each study site will facilitate each focus group, each have experience in undertaking qualitative research and will also take observational notes. A focus group topic guide will be used to structure the focus group. This will be informed by the data collected in stage 1 and qualitative data collected in stage 2 male remand prisoner interviews. The focus groups will last approximately 60 min.

*Data entry and analysis*: Focus groups will be digitally recorded, transcribed and analysed as per stage 2 male remand prisoner interviews. We will also be cognisant of the particular nuances of focus group data analysis and the need to focus on the intention and purpose of the study to ensure that we can make sense of the large amounts of data that can be generated.[46]

### Stage 3
#### Adaptation of intervention
Intervention mapping will be used to refine and develop an existing self-efficacy enhancing ABI[47] to reduce reported levels of alcohol consumption in male remand prisoners. Using an existing self-efficacy enhancing ABI framework, detailed mapping will be undertaken with an intervention specification developed, matrix of objectives and determinants identified and an implementation strategy produced. Members of the research study team will undertake this activity with input from the advisory group.

#### Ethical considerations
Multisite ethical approval was sought through the Integrated Research Application System (IRAS), and ethical approval was obtained from NRES, NOMS, R&D SPS, School of Health in Social Science, The University of Edinburgh. It is also important to ensure that the researchers are fully supported and cognisant of the issues involved in working within a prison environment. To address this, they will undertake prison delivered training, which will educate them on relevant protocols and risk assessment tools that the prisons use. The training is typical of that given to staff working in the prisons and includes for example personal protection training, fire safety training, hostage training, professional boundaries and suicide awareness and prevention. They will be provided with a personal alarm, will be accompanied by staff wherever necessary and will complete university risk assessments. During fieldwork, the researchers will have regular debriefing sessions with the PI (AH) and coinvestigator (DNB), will speak to each other weekly and will keep research diaries.

#### Consent
For this study, we consider valid consent to be underpinned by adequate information being provided to the potential study participant, and that they have the

capacity to decide whether or not they want to take part. Drawing from the Royal College of Nursing guidance on informed consent in health and social care research,[48] a capable person is defined as one who will:

► understand the purpose and nature of the research
► understand what the research involves, its benefits (or lack of benefits), risks and burdens
► understand the alternatives to taking part
► be able to retain the information long enough to make an effective decision
► be able to make a free choice
► be capable of making *this particular decision* at the time it needs to be made.

Freedom of consent can easily be undermined for prisoners and those in the criminal justice system, which means they may be more vulnerable to exploitation or abuse by researchers. For example, learning disabilities, illiteracy and language barriers are prevalent within these populations (www.prisonreformtrust.org.uk). Alongside the power differential between researcher and potential participant, particular care is needed to ensure that valid, freely given and fully informed consent can be achieved. We will train the researchers to use the Offender Health Research Network Toolkit that outlines a pathway to successfully undertake health research in the criminal justice system (www.ohrn.nhs.uk/toolkit/). The expert advisory group will also provide support, while professional codes of ethics (Nursing and Midwifery Council) will guide the team to safeguard the civil rights of subjects. The team has experience of undertaking research with offenders and in criminal justice settings, for example, prisons, probation and addiction services.

### Monitoring

We have convened an advisory group with terms of reference. The group will meet three times during the duration of the 18-month study to review progress of the project and to advise on engagement and dissemination.

### Data management

To optimise the security of our data, a database will be held at each university site. All data will be treated confidentially and stored securely and anonymously. Datasets will be created and maintained separate to participants' non-identifiable research data, and linked using unique identifier code, during collection, storage, management and transfer processes. All data will be accessible to project staff only. This system will be used for both hard copy and electronic files, for example, questionnaires, interview schedules, audio recordings, transcripts and database records. Any transporting or transmitting of data will ensure that personal/sensitive and wider data are transported separately to each other and in a secure manner. This will include transport from fieldwork sites to the research head office at The University of Edinburgh, or electronic transmission, if required. No personal data will be transferred outside the borders of the UK, or stored or collected on computer servers outside of UK borders. Any requirements to pass any personal data to another organisation will be approved by NHS Health Scotland in advance. The University of Edinburgh complies with the Data Protection Act 1998 and the University has a Digital Curation Centre that provides support and advice regarding data management planning (DMP) for researchers within the University. This enables researchers to undertake DMP according to the requirements stipulated by the major UK funders. Accordingly, we will work with the centre to build our DMP to ensure that we meet MRC requirements with regards to research integrity and replication, ensuring research data and records are accurate, complete, authentic and reliable alongside increasing research efficiency, saving time and resources in the long term while enhancing data security and minimising the risk of data loss.

### Dissemination

Our dissemination plan includes local, national and international communication, and the dissemination strategy will be a key output of the advisory group in collaboration with the research team. We will ensure that digital and media communication are utilised as part of the strategy. We will publish a full account of our research through open access peer-reviewed journal articles. Our outputs will be recorded on Researchfish. Findings of the study and the proposed pilot RCT will be disseminated to Health and Justice Teams at Scottish Government and Public Health England, the National Prisoner Healthcare Network (Scotland) and the WHO (Health in Prisons Programme Collaborating Centre). We will present our research at meetings/workshops/events of appropriate learned societies, for example, Scottish Alcohol Research Network, Scottish Health Action on Alcohol Problems and Offender Health Research Network. In addition, we will present our findings at national and international conferences. We will work with press officers at The University of Edinburgh to publicise the results of our work to local, national and international news media including radio.

**Acknowledgements** The study is sponsored by The University of Edinburgh. The authors would like to thank Andreana Adamson (NHS Director of Health and Justice, Scottish Government), Kieran Lynch (Criminal Justice Programme Manager, Public Health England), Tom Byrne (National Prisons Pharmacy Adviser, Health Improvement Scotland), John Porter (Prison Healthcare Lead Nurse, Health Improvement Scotland), Phil Eaglesham (Organisational Lead for Health Equity (Community Justice), NHS Health Scotland), Steven McCann and Stuart Wright (Scottish Prison Service), Tim Allen and Craig Phipps (English Prison Service) for their support of the study. We would also like to thank prison staff and peer support prisoners for their continued support during planning and data collection. Our thanks also to the study advisory group for their advice and support. Finally, our greatest thanks to all the prisoners and prison stakeholders who gave up their time to participate.

**Contributors** AH and SL wrote the first draft of the paper. DN-B commented on the first draft. AH, DN-B and AS contributed to the inception of the study. AH, DN-B, AS, PS, RP and JF contributed to the study design and development. AH managed the overall study, AH and DN-B coordinated the Scottish and English study sites, respectively. SL and JF led the quantitative and qualitative data collection. All coauthors contributed to the revisions and final draft of the paper.

**Competing interests** None declared.

**Patient consent** Participants in the study were not recruited as NHS patients, they were recruited as individuals incarcerated within Her Majesty's Prison (HMP) establishments. All participants were provided with a study information leaflet and

signed a consent form that made them aware that the findings would be used in peer-reviewed journals.

**Ethics approval** Ethical approval was given by the REC: South East Scotland Research Ethics Committee 01 (REC ref no 16/SS/0009). Scottish Prison Service (SPS), The University of Edinburgh School of Health in Social Science Ethics Committee, National Offender Management System (NOMS) and NHS Lothian Health Board Research & Development (R&D).

**Provenance and peer review** Not commissioned; externally peer reviewed.

**Open Access** This is an Open Access article distributed in accordance with the terms of the Creative Commons Attribution (CC BY 4.0) license, which permits others to distribute, remix, adapt and build upon this work, for commercial use, provided the original work is properly cited. See: http://creativecommons.org/licenses/by/4.0/

© Article author(s) (or their employer(s) unless otherwise stated in the text of the article) 2017. All rights reserved. No commercial use is permitted unless otherwise expressly granted.

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
