## [Reviewer comments · BMJ Open]

ARTICLE DETAILS

TITLE (PROVISIONAL)	Alcohol Brief Interventions for male remand prisoners: Protocol for a complex intervention framework development and feasibility study (PRISM-A)
AUTHORS	Holloway, Aisha; Landale, Sarah; Ferguson, Jennifer; Newbury-Birch, Dorothy; Parker, Richard; Smith, Pam; Sheikh, Aziz

VERSION 1 - REVIEW

REVIEWER	Joseph Barry Trinity College, Dublin, Ireland
REVIEW RETURNED	30-Oct-2016

GENERAL COMMENTS	Two categories of change are recommended. Question 12 of checklist. The study is only being conducted in two prisons. The authors need to comment on any limitations that may arise from their results. What if the results from the two sites are radically different? Question 15 of checklist. There are a number of minor typographical errors and editing anomalies that need to be corrected. These include; a) the title specifies remand prisoners but non-remand prisoners are mentioned in stage 1 but not again. If the non-remand prisoner results are relevant then non-remand prisoners need to be included in the title. If not relevant, then remove reference to non-remand prisoners in stage 1 b) the manuscript would be improved if an English language scholar proof read it. Among the issues are: in the introduction, 2nd line 'alcohol attributed' is not the appropriate phrase; the possessive apostrophe is incorrectly used in several places (1st paragraph of the introduction, 3rd bulleted objective); incorrect prepositions, tenses and other minor errors such as plurals for singulars and vice versa and numerous places that require a space bar c) Rules 41 and 95 on page 7 need to be explained d) on page 15 there is an asterisk after the word 'researcher' with no further explanation
--

REVIEWER	Michael Martin University of Ottawa, Canada
REVIEW RETURNED	22-Nov-2016

GENERAL COMMENTS	Thank you for the opportunity to review the protocol 'Alcohol Brief Interventions for male remand prisoners: Protocol for a complex intervention framework development and feasibility study (PRISM-A)'. Given that I understand data collection to be nearing completion (i.e. by December 2016), I have not commented on the methods in detail. While I believe the study design to be reasonably sound, I note a few comments below about the protocol for further refinement.  - It is unclear why convicted prisoners are sampled in phase 1, but not in phase 2. If the gap in the literature regarding ABI applies only to remand prisoners, then it would seem the study design should focus exclusively on this group. If ABI needs to be tailored to all prisoners, then excluding convicted prisoners seems to neglect their views in the development of such an intervention. While it may be late to change the design in light of data collection dates in the protocol, it seems that from an ethical perspective the balance of benefit (i.e. estimating prevalence with a screening tool rather than a gold standard diagnosis) to harm (i.e. time to participate, etc.) is minimal (in particular for the convicted group) given that the prevalence and burden of substance abuse is well documented among prison populations. If changes to the design remain possible, it would seem that a greater investment in the feasibility portion among convicted prisoners would offer a greater benefit to harm ratio. - It is unclear how much quantitative data is being collected, and how this will be analyzed for the analyses regarding feasibility and acceptability of an ABI. - It sounds as though the authors are taking a stratified sample to ensure sufficient numbers of remand prisoners. There is no mention of how the unequal sampling probabilities will be accounted for in the quantitative analyses. There is also no mention of power calculations to justify the overall and sub-group sample sizes.
---

REVIEWER	Geoff Page Mental Health and Addictions Research Group, Department of Health Sciences, University of York, UK
REVIEW RETURNED	28-Nov-2016

GENERAL COMMENTS	Overall, this is a strong and well-designed protocol for a study intended to assess the prevalence of alcohol use disorders within two prisons, and to assess the suitability of an alcohol brief intervention (ABI) for remand populations. Insofar as I have concerns about this protocol, they are relatively minor, and should be easy to address. The review of literature on alcohol-related crime is neither particularly robust, nor particularly well-referenced. This is clearly not the primary focus of this paper; but may benefit from some additional attention if alcohol-related crime (and associated savings) are to be a substantial consideration in future publications. Remand populations may continue to serve a full prison sentence, or
--

	be returned to the community. Some exploration (or acknowledgment) of the significance of this for an ABI focused on remand populations would be a positive addition. The planned sample size is not clearly explained. The planned statistical analyses, as described, are entirely descriptive, raising questions about why interviewing 500 prisoners comprises an efficient use of resources. Moreover, attempting to interview 250 prisoners in each site looks highly ambitious, and highly resource intensive for both research associates and prison staff. Even when working in highly supportive prisons, with full support from governors and frontline staff, prisoners are only likely to be available for – at most – a handful of hours each day. Difficulties in locating prisoners (who may have unexpectedly been released or transferred to another wing; gone to work or education; or stayed away from work or education due to unexpected illness, legal visits, or other reasons), unexpected lockdowns, and a lack of available escort staff can, in our recent experience, further compound the difficulties of locating and interviewing named individuals. The time taken to transport prisoners from assorted locations to a private meeting room would further extend any timeframes. In sum, securing face-to-face interviews with 500 prisoners in confidential offices or rooms looks like a highly ambitious goal. If the authors are confident that this is achievable, some information on projected timeframes for securing the full sample and / or the processes they have employed would support future replication. Recruitment as described (p.9, lines 54-58) appears to rely on written information. This will, presumably, exclude prisoners with literacy problems. This should be acknowledged, as should any likely impact on recruitment and the study's findings. There is some repetition within the aims – it is not clear how 'understand[ing] the feasibility and acceptability amongst adult male remand prisoners of an ABI designed for them' differs from 'explor[ing] adult male remand prisoners' ... views regarding the acceptability of receiving an ABI whilst on remand' If these relatively minor points are addressed, I have little hesitation in recommending this paper for publication.
--	---

VERSION 1 – AUTHOR RESPONSE

Reviewer Joseph Barry (1)

2 Question 12 of checklist. The study is only being conducted in two prisons. The authors need to comment on any limitations that may arise from their results. What if the results from the two sites are radically different?

Response: We acknowledge the risk of only having two prison sites impacting on representativeness. We are also aware that there are many regimes within the prison estate across the UK and many prisons with a range of prisoner categories. In order that we gain a fuller insight into the variety for the purposes of a future RCT, we sought guidance from colleagues in criminal justice, NOMS and SPS to identify two appropriate prisons. We had added this information to Pg 9 under section 'Participants and setting'.

3 There are a number of minor typographical errors and editing anomalies that need to be corrected.

Response: Typographical errors and editing anomalies corrected.

4 The title specifies remand prisoners but non-remand prisoners are mentioned in stage 1 but not

again. If the non-remand prisoner results are relevant then non-remand prisoners need to be included in the title. If not relevant, then remove reference to non-remand prisoners in stage 1

Response: Remand prisoners and non-remand were the focus of Phase one. Phase one was specifically designed so that we could look at the prevalence rates between remand and convicted prisoners. Non-remand were not the focus of Phase two. At present we are working on developing brief intervention with remand prisoners, but the prevalence information gathered in phase one is imperative to the field as a whole. The levels of substance use using the AUDIT (as the gold standard screening tool) have not been well measured in the UK. We have kept the original title. Newbury-Birch D, McGovern R, Birch J, O'Neill G, Kaner H, Sondhi A, Lynch K: A rapid systematic review of what we know about alcohol use disorders and brief interventions in the criminal justice system. *International Journal of Prisoner Health* 2016, 12(1):57-70.

5 The manuscript would be improved if an English language scholar proof read it. Among the issues are: in the introduction, 2nd line 'alcohol attributed' is not the appropriate phrase; the possessive apostrophe is incorrectly used in several places (1st paragraph of the introduction, 3rd bulleted objective); incorrect prepositions, tenses and other minor errors such as plurals for singulars and vice versa and numerous places that require a space bar

Response: Manuscript proof read and corrected.

6 Rules 41 and 95 on page 7 need to be explained

Response: Explanation of what the rules are and reference to the document *The Prisons and Young Offenders Institutions (Scotland) Rules 2011* have been added to pg 10.

7 On page 15 there is an asterix after the word 'researcher' with no further

Response: Asterix deleted (typo).

Reviewer Michael Martin (2)

8 It is unclear why convicted prisoners are sampled in phase 1, but not in phase 2. If the gap in the literature regarding ABI applies only to remand prisoners, then it would seem the study design should focus exclusively on this group. If ABI needs to be tailored to all prisoners, then excluding convicted prisoners seems to neglect their views in the development of such an intervention. While it may be late to change the design in light of data collection dates in the protocol, it seems that from an ethical perspective the balance of benefit (i.e. estimating prevalence with a screening tool rather than a gold standard diagnosis) to harm (i.e. time to participate, etc.) is minimal (in particular for the convicted group) given that the prevalence and burden of substance abuse is well documented among prison populations. If changes to the design remain possible, it would seem that a greater investment in the feasibility portion among convicted prisoners would offer a greater benefit to harm ratio.

Response: Phase one was specifically designed so that we could look at the prevalence rates between remand and convicted prisoners. At present we are working on developing brief intervention with remand prisoners but this information is imperative to the field. The levels of substance use using the AUDIT (as the gold standard screening tool) have not been well measured in the UK.

Newbury-Birch D, McGovern R, Birch J, O'Neill G, Kaner H, Sondhi A, Lynch K: A rapid systematic review of what we know about alcohol use disorders and brief interventions in the criminal justice system. *International Journal of Prisoner Health* 2016, 12(1):57-70.

9 It is unclear how much quantitative data is being collected, and how this will be analyzed for the analyses regarding feasibility and acceptability of an ABI.

Response: On page 11 of the manuscript, under the heading "Study design" we currently state that: "We will conduct an interviewer led cross-sectional survey, delivered by the study RAs. Based on previous studies we anticipate recruiting 500 participants (n=250 at each site) of that approximately 100 (n=50 at each site) will be male remand prisoners." Further down on page 12 of the paper (under the heading "Data entry and analysis") we write that "Descriptive statistics and 95% confidence intervals will be used to summarize the data and inform the design of a potential future pilot trial. Variables will be classified and described using frequency tables, mean, median and standard deviation. The prevalence of drinking at harmful, hazardous and dependent levels and corresponding 95% confidence intervals will be calculated."

10 It sounds as though the authors are taking a stratified sample to ensure sufficient numbers of remand prisoners. There is no mention of how the unequal sampling probabilities will be accounted for in the quantitative analyses.

Response: Thank you for your comment. We will take the stratified data collection into account by performing a stratified analysis, which will involve performing the analysis in the remand and non-remand prisoner groups separately, and therefore we do not need to take into account unequal sampling probabilities. We have now clarified in the manuscript that analysis will be conducted in the remand and non-remand prisoner groups separately (page 12): "The quantitative analysis will be stratified such that the analysis will be conducted in the remand and non-remand prisoner groups separately."

11 There is also no mention of power calculations to justify the overall and sub-group sample sizes.

Response: The sample size justification is based on the precision around 95% confidence intervals, as is commonly done in feasibility studies. A sample size of 100 remand prisoners ensures that the half-width of a 2-sided 95% confidence interval for a continuous outcome (e.g. the total AUDIT score) is no more than 0.2 times the standard deviation; and the half-width of a 2-sided 95% confidence interval for a proportion (e.g. the prevalence of drinking at harmful, hazardous, and dependent levels) is no more than 10%. A sample size of 400 convicted prisoners ensures that the half-width of a 2-sided 95% confidence interval for a continuous outcome (e.g. the total AUDIT score) is no more than 0.1 times the standard deviation; and the half-width of a 2-sided 95% confidence interval for a proportion (e.g. the prevalence of drinking at harmful, hazardous, and dependent levels) is no more than 5%.

Reviewer Geoff Page (3)

12 The review of literature on alcohol-related crime is neither particularly robust, nor particularly well-referenced. This is clearly not the primary focus of this paper; but may benefit from some additional attention if alcohol-related crime (and associated savings) are to be a substantial consideration in future publications.

Reviewer: We agree with the reviewer that alcohol-related crime is not the primary focus, however we have added two paragraphs to expand on this area with relevant citations (pg 6).

13 Remand populations may continue to serve a full prison sentence, or be returned to the community. Some exploration (or acknowledgment) of the significance of this for an ABI focused on remand populations would be a positive addition.

Responses: The specific objectives Pg 7 identify that we will explore the perspectives of male remand prisoners in relation to intervention delivery points, techniques, methods of delivery and by whom.

This will address the comments from the reviewer who rightly identifies that remand prisoners may stay in prison or return to the community. Therefore we will explore this from both perspectives.

14 The planned sample size is not clearly explained.

Response: See response to Reviewer 2 regarding same issue

15 The planned statistical analyses, as described, are entirely descriptive, raising questions about why interviewing 500 prisoners comprises an efficient use of resources.

Response: See above sample size justification. We believe that interviewing 500 prisoners is an efficient and worthwhile use of resources at this feasibility stage in order that we can fully explore the likelihood of successfully recruiting to a future multi centre RCT. There is very little prior data available on this and very little work done in this area. We have clarified in the paper that confidence intervals as well as descriptive statistics will be used to summarise the data (p11): "Descriptive statistics and 95% confidence intervals will be used to summarise the data and inform the design of a potential future pilot trial." Performing a long list of formal hypothesis/significance tests would not be appropriate in this context because this is feasibility study and we only wish to inform the design of a future larger study. Lancaster et al. [ref] recommend that the analysis of a pilot or feasibility study should be mainly descriptive and focus on confidence interval estimation.

Lancaster, Gillian A., Susanna Dodd, and Paula R. Williamson. "Design and analysis of pilot studies: recommendations for good practice." *Journal of evaluation in clinical practice* 10.2 (2004): 307-312.

16 Moreover, attempting to interview 250 prisoners in each site looks highly ambitious, and highly resource intensive for both research associates and prison staff. Even when working in highly supportive prisons, with full support from governors and frontline staff, prisoners are only likely to be available for – at most – a handful of hours each day. Difficulties in locating prisoners (who may have unexpectedly been released or transferred to another wing; gone to work or education; or stayed away from work or education due to unexpected illness, legal visits, or other reasons), unexpected lockdowns, and a lack of available escort staff can, in our recent experience, further compound the difficulties of locating and interviewing named individuals. The time taken to transport prisoners from assorted locations to a private meeting room would further extend any timeframes. In sum, securing face-to-face interviews with 500 prisoners in confidential offices or rooms looks like a highly ambitious goal. If the authors are confident that this is achievable, some information on projected timeframes for securing the full sample and / or the processes they have employed would support future replication.

Response: The reviewer has identified an important issue with regards collecting data in this setting and population for all the reasons they have provided. We have just completed prisoner data collection and have exceeded the 250 at each site, with over 500 in total. We have added a small section "Prison data collection preparation" on Pg 9 with some additional information and timescales as suggested.

17 Recruitment as described (p.9, lines 54-58) appears to rely on written information. This will, presumably, exclude prisoners with literacy problems. This should be acknowledged, as should any likely impact on recruitment and the study's findings.

Response: This is a very important point. We engaged with Service users through a charity organisation when developing the study information leaflet and consent form. Whilst we tried to

ensure the language was free from jargon and appropriate for the intended audience for ethics purposes there was a need to include certain information. We will record the number of exclusions due to literacy so we can identify the number affected by this. For follow on studies we will look to develop innovative visual methods to engage (DN-B co-author and co-ordinator for one of the study sites has had experience of working in this way in the past with school children). Sentences added to identify that we will record incidences of where this has happened (pg 10) and engagement with service users in development of study materials.

18 There is some repetition within the aims – it is not clear how ‘understand[ing] the feasibility and acceptability amongst adult male remand prisoners of an ABI designed for them’ differs from ‘explor[ing] adult male remand prisoners’ ... views regarding the acceptability of receiving an ABI whilst on remand’

Response: The second objective has been deleted to avoid repetition
 ‘understand[ing] the feasibility and acceptability amongst adult male remand prisoners of an ABI designed for them’

VERSION 2 – REVIEW

REVIEWER	Geoff Page Mental Health and Addictions Research Group, Department of Health Sciences, University of York
REVIEW RETURNED	18-Jan-2017

GENERAL COMMENTS	The research team are to be commended on securing >500 initial / face-to-face interviews already; this is an impressive achievement. Both new paragraphs on page 6 would benefit from a proof read; neither is entirely grammatical. The additional paragraph on p.10 needs to be revisited - 'The Information Sheet and the study consent form will be Research Assistant (RAs) will obtain...' My point about remand populations may not have been clear, as it does not appear to be answered within the study's objectives (nor does this appear to be a particularly suitable place to answer it). Interventions delivered within remand are in a curious position - approximately 25% of those who receive a remand-based ABI will be acquitted or receive a non-custodial sentence, and will return to the community imminently. This group will be surrounded by alcohol (and alcohol-infused peers, temptation, etc), and will have an immediate opportunity to put their ABI learning into practice. Indeed, this immediacy is part of the reason that 'through the gate' interventions (both psychosocial and pragmatic) are seen as valuable, and have been an increasing focus of UK policy and practice. The other 75% of remandees receive prison sentences, some of which may be for many years. For this group, whatever the other challenges and temptations, widespread alcohol use and the potential to drink at hazardous levels will not exist for the duration of their imprisonment. This appears to be an important factor affecting almost every aspect of remand-based interventions - unless the skills / tools / lessons learnt in an ABI can be reasonably expected to be held in stasis for the duration of a (potentially lengthy) prison sentence (and this, in and of itself, appears to be a bold assumption that would benefit
--

	from some coverage - even if brief). Prisoner interviewees, as the authors identify on p.7, will no doubt have some say about how they feel interventions could consequently be planned ('delivery points, techniques, methods of delivery, and by whom' in the comments). However, this does not feel like a substantive response to the unique context of remand, or the significance of ABI recipients returning to such very different contexts (prison / community).
--	---

VERSION 2 – AUTHOR RESPONSE

Geoff Page

1

Both new paragraphs on page 6 would benefit from a proof read; neither is entirely grammatical.
Response: Both paragraphs have been proof read and edits made.

2

The additional paragraph on p.10 needs to be revisited - 'The Information Sheet and the study consent form will be Research Assistant (RAs) will obtain...'

Response: Additional paragraph on p. 10 has been amended and edited.

3

My point about remand populations may not have been clear, as it does not appear to be answered within the study's objectives (nor does this appear to be a particularly suitable place to answer it). Interventions delivered within remand are in a curious position - approximately 25% of those who receive a remand-based ABI will be acquitted or receive a non-custodial sentence, and will return to the community imminently. This group will be surrounded by alcohol (and alcohol-infused peers, temptation, etc), and will have an immediate opportunity to put their ABI learning into practice. Indeed, this immediacy is part of the reason that 'through the gate' interventions (both psychosocial and pragmatic) are seen as valuable, and have been an increasing focus of UK policy and practice.

The other 75% of remandees receive prison sentences, some of which may be for many years. For this group, whatever the other challenges and temptations, widespread alcohol use and the potential to drink at hazardous levels will not exist for the duration of their imprisonment.

This appears to be an important factor affecting almost every aspect of remand-based interventions - unless the skills / tools / lessons learnt in an ABI can be reasonably expected to be held in stasis for the duration of a (potentially lengthy) prison sentence (and this, in and of itself, appears to be a bold assumption that would benefit from some coverage - even if brief).

Prisoner interviewees, as the authors identify on p.7, will no doubt have some say about how they feel interventions could consequently be planned ('delivery points, techniques, methods of delivery, and by whom' in the comments). However, this does not feel like a substantive response to the unique context of remand, or the significance of ABI recipients returning to such very different contexts (prison / community).

Response: The reviewer makes a very important point here with regards the delivery of interventions such as ABIs for remand prisoners of which a smaller proportion will be liberated in comparison to those who will receive prison sentences.

We have gone back to the original reviewer point from original submission made (13) “Remand populations may continue to serve a full prison sentence, or be returned to the community. Some exploration (or acknowledgment) of the significance of this for an ABI focused on remand populations would be a positive addition”

In response therefore, we have included a paragraph on p.6 to acknowledge the significance of an ABI for remand populations who may or may not be liberated, as suggested by the reviewer.

VERSION 3 – REVIEW

REVIEWER	Geoff Page University of York, UK
REVIEW RETURNED	21-Mar-2017

GENERAL COMMENTS	As noted before, this is a solid paper outlining the protocol for a study that explores the viability of an ABI in a much-missed population. With the changes made, I have no hesitation in recommending it for publication. (Though I would suggest that the authors fact-check my earlier figure for the proportion of remandees who progress to imprisonment, as I believe they have taken my figure as gospel, when my search may have been a bit overly hasty).
--